# Pore Structure Characteristics and Strength Variation of Red Sandstone under Freeze–Thaw Cycles

**DOI:** 10.3390/ma15113856

**Published:** 2022-05-28

**Authors:** Yongwei Lan, Hongmei Gao, Yanlin Zhao

**Affiliations:** 1College of Mining Engineering, Heilongjiang University of Science and Technology, Harbin 150022, China; 2005800164@usth.edu.cn; 2School of Architecture and Civil Engineering, Heilongjiang University of Science and Technology, Harbin 150022, China; 1995801017@usth.edu.cn

**Keywords:** freeze–thaw cycles, uniaxial compressive strength, microscopic pore structure, porosity

## Abstract

To study pore structure characteristics and the strength of red sandstone under freeze–thaw cycles, the saturated red sandstone was studied by the combination of freeze–thaw cycle test, high-pressure mercury injection test, uniaxial compression test and theoretical analysis, and research shows that: with the increase of freeze–thaw cycles, the pores of red sandstone continue to expand and extend, macropore volume and the total pore volume increases gradually, and the pore size distribution curves become more continuous. Porosity of samples after 10, 30, 70 and 100 freeze–thaw cycles is 1.14 times, 1.17 times, 1.28 times and 1.44 times of that of 0 cycle, and the uniaxial compressive strength of samples is 0.68 times, 0.53 times, 0.26 times and 0.17 times of that of 0 cycle, respectively. With the increase of freeze–thaw cycles, freeze–thaw damage continues to accumulate, the crack propagation direction changes from axial through-through failure mode to transverse and axial simultaneous failure mode. Taken the change of porosity as a parameter, through the regression analysis of the test data, the functional relationship between uniaxial compressive strength and the change of porosity in red sandstone is established. The research results will provide a theoretical basis for stability research of slopes of railway subgrade in cold region.

## 1. Introduction

The quantity and scale of geotechnical engineering in cold regions of China are relatively large. Freeze–thaw cycles pose a great threat to large-scale rock slope projects in cold regions [1,2]. Therefore, it is of important engineering value to explore the degradation mechanism of mechanical properties in rock under freeze–thaw cycles.

Experimental and theoretical efforts have been made around the world to investigate the variation of physico-mechanical properties in rocks with freeze–thaw cycles. Scholars at home and abroad [3,4,5,6,7,8,9,10,11,12] obtained the variations in saturated water absorption, longitudinal wave velocity, density, Poisson’s ratio, elastic modulus, rate of swelling-shrinkage, internal friction angle, uniaxial compressive strength, triaxial compressive strength, frost heave force, freeze–thaw coefficient, fracture toughness, energy release rate and creep characteristics of rocks after freeze–thaw cycles. Khanlari [9], Zhou [11] and Wu [12] obtained the relationship between strength and the number of freeze–thaw cycles. Jia [13], Han [14], Li [15] and Zhang [16] analyzed the freeze–thaw damage mechanism in rock with different freezing conditions. Huang [17,18] studied the mechanism of rock crack propagation after freeze–thaw cycles. Niu [19,20], Yang [21] studied the dynamic mechanical properties of rock and determined the variation law of dynamic strength of rock with freeze–thaw cycle. Li [22], Gao [23], Huang [24] and Zhai [25] researched the important effect of pore development on mechanical damage in rock. Yu [26] explored the porosity, compressive strength and microstructure of limestone samples before and after freeze–thaw cycle.

Xu [27], Gao [28] and Pei [29] studied the variation law of strength of water saturated sandstone under the action of freeze–thaw. Jia [30], Rong [31] and Huang [32] studied the variation law of porosity of water saturated sandstone under the action of freeze–thaw.

However, there are few quantitative research results on the pore structure characteristics [27,28,29], the variation law of compressive strength [30,31,32] and the relationship between porosity and strength of saturated red sandstone under freeze–thaw cycle is not seen. Thus, the variation of micro-pore structure and strength of sandstone with freeze–thaw cycles still needs to be deeply studied. Taken red sandstone as the research object, the pore structure characteristics and the change law of uniaxial compressive strength of red sandstone under freeze–thaw cycles are studied, and the relationship between strength and porosity of red sandstone under freeze–thaw cycles is explored. The research results will provide a theoretical basis for stability research of slopes of railway subgrade in cold region.

## 2. Test Overview

### 2.1. Samples of Red Sandstone

Red sandstone from rock slope engineering in Rizhao city, Shandong Province, was sampled by core drilling. According to the sample preparation requirements of the International Society for Rock Mechanics [6,27] (ISRM), standard samples should be prepared such that each had a diameter of 50 mm and height of 100 mm, flatness of less than 0.05 mm on both ends, and perpendicularity of less than 0.20°. Visual inspection should be performed on the prepared samples to identify and exclude those with evident surface defects. Samples with good uniformity and consistent texture were selected.

### 2.2. Test Instruments

The freeze–thaw test chamber is shown in Figure 1, which is composed of four systems: heating, cooling, temperature control, and temperature recording. The test chamber also has a dual-limit temperature control function and looped freeze–thaw treatment. The temperature is controlled by a thermistor sensor with accuracy of ±0.2 °C. The range of temperature variation is displayed and recorded by the temperature recording system.

The mercury injection instrument is seen in Figure 2. Porosity, the total pore volume and transmissibility were determined using the mercury intrusion method. The mercury injection instrument is equipped with two low-pressure stations and one high-pressure station, and volume accuracy of mercury entering or exiting mercury is better than 0.1 μL. The system also features a powerful built-in data processing and reporting program package, measuring porosity with accuracy of ±0.001%.

Rock Triaxial Test System with an electro-hydraulic servo is shown in Figure 3, mainly consists of a power oil source, an axial loading frame, a dynamic confining pressure control system, and an all-digital control system, which is used for uniaxial loading rock mechanical damage tests. When loading, the axial displacement rate is used to control the displacement rate at 0.001 mm/s, and the axial extensometer is used to test the axial deformation of the samples.

### 2.3. Test Design

The red sandstone samples were soaked in water at 20 °C for 2 days to make sandstone saturated. The saturated samples were subjected to 0, 10, 30, 70, and 100 freeze–thaw cycles, respectively. Each freeze–thaw cycle was as follows: the temperature started from +20 °C, slowly decreased to the specified temperature (−20 °C), maintained the constant temperature for 8 h, then increased to +20 °C, kept the constant temperature for 4 h, and temperature recording system automatically controlled the time. One freeze–thaw cycle was about 12 h. The samples after freeze–thaw are shown in Figure 4. The density of red sandstone is 2.3429 g/cm^3^, and the main mineral components of red sandstone are quartz, albite, plagioclase and illite. Mercury injection test [22] and uniaxial compression test [27] were carried out on samples.

## 3. Analysis of Experimental Results

### 3.1. Mercury Injection Test Results

According to the decimal rock pore classification standard [30,31], the pores in red sandstone can be divided into four types: macropores (diameter > 1000 nm), mesopores (1000 nm ≥ diameter > 100 nm), transitional-pores (100 nm ≥ diameter > 10 nm) and micropores (diameter ≤ 10 nm).

#### 3.1.1. Pore Volume Versus Pore Diameter Curves

As expressed in Figure 5, curves of pore volume and pore diameter of red sandstone are “S shapes”. In the micropores section, the pore volume curves are relatively flat, indicating that the micropore volumes do not change significantly. In transitional-pores and mesopores section, the change curves of pore volume are steep, indicating that the pore volume of transitional-pore and mesopores changes greatly. In the macropores section, the change curves of pore volume are the steepest, indicating that macropores increment are the largest. With the increasing of freeze–thaw cycles, pore volume of red sandstone gradually increases and the continuity of pore volume curves are enhanced, it reflects that the pores in sandstone are gradually developed, the pores connectivity are improved, and the freeze–thaw damages accumulate continuously.

#### 3.1.2. Pore Volume Distribution of Red Sandstone

Pore volume parameters of samples are shown in Table 1, and pore volume distribution is shown in Figure 6.

As can be seen from Table 1 and Figure 6:

Compared with 0 cycle, macropores volume of red sandstone increases, mesopores volume decreases, micropores volume changes less, and the total pore volume increases gradually after 10 and 30 cycles. It is because the frost heaving force generated by freeze–thaw cycle changes the cohesion of mineral particles in sandstone, and the weak surfaces of mineral particle cementation continue to weaken, some micropores, transitional-pores and mesopores of red sandstone are transformed into macropores. The connectivity of macropores is gradually enhanced, and the damage of red sandstone is continuously accumulated.

Compared with 30 cycles, transitional-pores volume and mesopore volume in red sandstone gradually increases, macropore volume slightly increases, micropore volume has little change, and the total pore volume gradually increases at 70 freeze–thaw cycles. This is because repeated freeze–thaw cycles continuously weaken the weak surfaces of mineral particle cementation, and the mineral particles fall off, and some macropores are filled with internal free micro particles and transform into medium pores or transition pores. Although the increasing trend of macropores volume is not obvious, the total pore volume continues to increase, and the internal damage of red sandstone continues to intensify.

Compared with 70 cycles, macropores volume and the total pore volume in red sandstone continue to increase at 100 cycles. This is because macropores are highly sensitive to freeze–thaw action, macropores develops rapidly, the proportion of macropores volume in the total pore volume is increasing constantly. Macroscopically freeze–thaw damage occurred in rock samples due to interpenetration and expansion of pores.

#### 3.1.3. Pore Size Distribution

Figure 7 shows that: When freeze–thaw are 0, 10, 30, 70 and 100 cycles, the pore diameters at the peak is 1510.5 nm, 2009.1 nm, 2330.7 nm, 7678.0 nm and 9821.4 nm, respectively. The pore size corresponding to the peak increases gradually, the peak moves to the right, and the average pore size increases gradually. As the increasing of freeze–thaw cycles, macropores show centralized distribution and the continuity of pore-size distribution curve is gradually enhanced. The pore connectivity increases, and the micro-damage of samples continues to accumulate.

#### 3.1.4. Porosity of Red Sandstone

Porosity of red sandstone is presented in Table 2, and the change curve of porosity is shown in Figure 8.

Table 2 and Figure 8 show that:

Average value of porosity in red sandstone is 11.6068% with 0 freeze–thaw cycle. Porosity of samples after 10 cycles is 1.14 times that of 0 freeze–thaw cycle. The corresponding increase of porosity per unit freeze–thaw cycle is largest, the slope of porosity is largest. At the initial stage of freeze–thaw, water in pores of red sandstone freezes into ice with large frost heave deformation and large frost heave force [17,18], which leads to continuous accumulation of damage on the weak surface of cementation between mineral particles and reduction of tensile strength of the weak surface between mineral particles. The initial pores expands, and new pores are formed. As a result, porosity increases dramatically.

Porosity of red sandstone after 30 cycles is 1.17 times that of 0 cycle, and porosity continued to increase. This is because the repeated action of frost heaving force makes the cohesion between mineral particles continue to weaken and the weak surface continues to deteriorate. The pores continue to expand and extend, and porosity continues to increase. Porosity of samples after 70 cycles is 1.28 times that of 0 cycle. It indicates that the repeated action of frost heaving-force continues to increase the damage in weak surface, the mineral particles of weak surfaces deteriorate and fall off, resulting in the continuous growth of porosity in red sandstone. Porosity of red sandstone after 100 cycles is 1.44 times that of 0 cycle. It shows that the pores in sandstone continues to develop, expand and connect. The freeze–thaw micro-damage of sandstone is added to macro-damage. That is consistent with results of Rong [30].

Variation law of porosity in red sandstone are consistent with the morphological characteristics of pore volume curve, pore volume distribution and pore size distribution.

### 3.2. Uniaxial Compression Test of Red Sandstone

#### 3.2.1. Uniaxial Compressive Strength

The results of uniaxial compression tests can be seen in Table 3. Typical stress–strain curves of samples are shown in Figure 9. The variation of uniaxial compressive strength is shown in Figure 10.

Figure 9 shows that the compaction section of stress–strain curves become more obvious with the increasing of freeze–thaw cycles. That is because the repeated frost heaving force reduces cohesion among particles of samples, and the damage of weak surfaces of the cementation among mineral particles continues to increase. The contact areas among mineral particles become smaller, porosity of red sandstone is increasing. The increase of porosity affect the deformation characteristics of red sandstone, and its plastic deformation ability also increases in different degrees.

Table 3, Figure 9 and Figure 10 show that the average value of the uniaxial compressive strength of red sandstone with 0 cycles is 40.423 MPa. The uniaxial compressive strength of samples after 10, 30, 70 and 100 freeze–thaw cycles is 0.68 times, 0.53 times, 0.26 times and 0.17 times of that of 0 cycle, respectively. The uniaxial compressive strength decreases with the increasing of freeze–thaw cycles, that is consistent with results of Pei [29].

This is mainly due to uneven deformation caused by the thermal expansion and the cold contraction of mineral particles and pore water in red sandstone, which leads to frost heaving force. The repeated actions of frost heaving force weaken the bonding degree of mineral particles and reduce the bonding force among particles, and the damage of weak surfaces of cementation among mineral particles is increasing. The falling off of mineral particles and the continuous development of internal pores lead to the continuous increase of macropores, porosity, average pore size, pore expansion and pore connectivity in red sandstone. As a result, the compressive strength of red sandstone continues to decrease. That is consistent with results of Rong [31].

Figure 10 shows the uniaxial compressive strength corresponding to the unit freeze–thaw cycles decreases very obviously from 0 to 10 cycles, and the slope in the strength change curve is the largest, which corresponds to the change law of porosity in Figure 8 (The corresponding increase of porosity with unit freeze–thaw cycles is largest, the change slope of porosity is largest.). It shows that the increment of porosity with per unit freeze–thaw cycles has a significant effect on the strength of samples.

#### 3.2.2. Failure Modes

The typical failure modes of red sandstone are shown in Figure 11. As can be found from the failure modes of samples in Figure 11a–c, there are obvious failure surfaces penetrating along the axial direction when the samples are broken. This is because the damage accumulation caused by freeze–thaw in the red sandstone is not enough to dominate the propagation direction of the main crack during failure before 70 cycles, although the strength of samples decreases continuously, failure modes are basically consistent with that of 0 cycle. It can be clearly seen from Figure 11d,e that there are failure surfaces perpendicular to the axial direction in the failure modes, and the failure mode is transverse and axial failure mode simultaneously. The analysis of the reason shows that the number of pores in red sandstone continues to accumulate and increase, a large number of pores are connected with each other, and macropores with similar sizes show centralized distribution with the increase of freeze–thaw cycles. The damage accumulation caused by freeze–thaw cycles in red sandstone can dominate the crack propagation trend, and the transverse failure severity of red sandstone increases.

## 4. Strength Prediction Model of Red Sandstone

Assuming that the porosity of red sandstone with 0 freeze–thaw cycle is $\Phi_{0}$, and the porosity after N cycles is Φ(N), which is assumed that Φ(N) is a function of the number of freeze–thaw times and is a differentiable function, the change of porosity from N cycles to (N+ΔN) cycles is
(1)Φ(N+ΔN)−Φ(N)=aΔN (a≠0)
where *a* is the change of porosity with per freeze–thaw cycle, and is a constant, as shown in Figure 8.

By integrating Equation (1), it can be expressed as:(2)$$\Phi \left(N\right)-\Phi_{0}=aN$$

Equation (2) can be further written as
(3)$$N=\frac{\Phi \left(N\right)-\Phi_{0}}{a}$$

Assuming that the peak strength of sample after N and (N+ΔN) cycles is $\sigma_{s}(N)$ and $\sigma_{s}(N+\Delta N)$, which is assumed that $\sigma_{s}(N)$ and $\sigma_{s}(N+\Delta N)$ are functions of the number of freeze–thaw times and are differentiable functions, the peak strength with 0 cycle is $\sigma_{s0}$, the loss rate of peak strength from N cycles to (N+ΔN) cycles is established
(4)$$\frac{\sigma_{s}\left(N\right)-\sigma_{s}\left(N+\Delta N\right)}{\sigma_{s}\left(N\right)}=b\Delta N\;(b\neq 0)$$
where b is the peak strength loss rate within the unit freeze–thaw cycle, and is a constant, as shown in Figure 10.

Equation (4) can be expressed as
(5)$$\frac{\sigma_{s}\left(N+\Delta N\right)-\sigma_{s}\left(N\right)}{\Delta N}=-b\sigma_{s}\left(N\right)$$

That is
(6)$$\frac{d\sigma_{s}\left(N\right)}{dN}=-b\sigma_{s}\left(N\right)$$

By integrating Equation (6), it can be expressed as
(7)$$\frac{\sigma_{s}\left(N\right)}{\sigma_{s0}{}_{}}=\exp\left(-bN\right)$$

Substituting Equation (3) into Equation (7), the following formula can be obtained
(8)$$\frac{\sigma_{s}\left(N\right)}{\sigma_{s0}}=\exp\left\{-\frac{a}{b}\left[\sigma_{s}\left(N\right)-\sigma_{s0}\right]\right\}$$

If $\Delta \Phi =\Phi (N)-\Phi_{0}$, Equation (8) can be written as
(9)$$\sigma_{s}\left(N\right)=c\sigma_{s0}\;\exp[-\frac{a}{b}\Delta \Phi ]$$where *c* is the peak strength correction coefficient within the number of freeze–thaw cycles. Initial conditions: when the freeze–thaw cycle is 0 cycle, the change rate of porosity is 0 and the uniaxial compressive strength is the test value of the test.

Equation (9) expresses that the relationship between the strength and the change of porosity in saturated red sandstone follows an exponential function under freeze–thaw cycles. Considering that initial peak strength and the changes of porosity do not completely obey a positive proportional function with freeze–thaw cycles, the above model needs to be fitted and modified by means of experimental data fitting.

Based on the above theory, the test data such as the initial strength and the change of porosity in red sandstone were selected for fitting. The fitting results are seen in Figure 12.

Figure 12 shows that the peak strength of red sandstone has a good correlation with the change of porosity. If the change of porosity is known, the corresponding uniaxial compressive strength can be predicted by Equation (9).

Based on the uniaxial compressive strength and porosity test data of limestone under freeze–thaw conditions in reference [26], the variation law of uniaxial compressive strength of limestone under freeze–thaw conditions is predicted according to Equation (9), as shown in Figure 13. Figure 13 shows that the predicted uniaxial compressive strength is basically close to the uniaxial compressive strength obtained from the test, indicating that the prediction model has certain value.

## 5. Conclusions

Taking red sandstone as the research object, high-pressure mercury injection test and the Uniaxial compression test of saturated red sandstone were carried out. The micro-pore structure and strength change law of saturated red sandstone after freeze–thaw cycles was studied, the following conclusions are drawn:(1)With the increase of freeze–thaw cycles, the total pore volume of red sandstone increases continuously, the continuity of pore size distribution curves increases, macropores show centralized distribution, and the freeze–thaw damage accumulates gradually. Porosity of samples after 10, 30, 70 and 100 freeze–thaw cycles is 1.14 times, 1.17 times, 1.28 times and 1.44 times of that of 0 cycle, and porosity increases obviously.(2)With the increase of freeze–thaw cycles, cohesion among mineral particles of red sandstone decreases, weak surfaces of cementation mineral particles deteriorate, the porosity increases obviously, the uniaxial compressive strength of samples after 10, 30, 70 and 100 freeze–thaw cycles is 0.68 times, 0.53 times, 0.26 times and 0.17 times of that of 0 cycle, respectively, the uniaxial compressive strength of samples decreases gradually, the failure modes change from the failure surface penetrating along the axial direction to the horizontal and longitudinal failure at the same time, and the transverse failure severity of red sandstone increases.(3)The change of porosity has a great influence on the uniaxial compressive strength of red sandstone. Taken the change of porosity as a parameter, the exponential relationship between the strength and the change of porosity in red sandstone is established. Through the regression analysis of test data, the strength prediction model of red sandstone after freeze–thaw cycles is modified.

## Figures and Tables

**Figure 1 materials-15-03856-f001:**
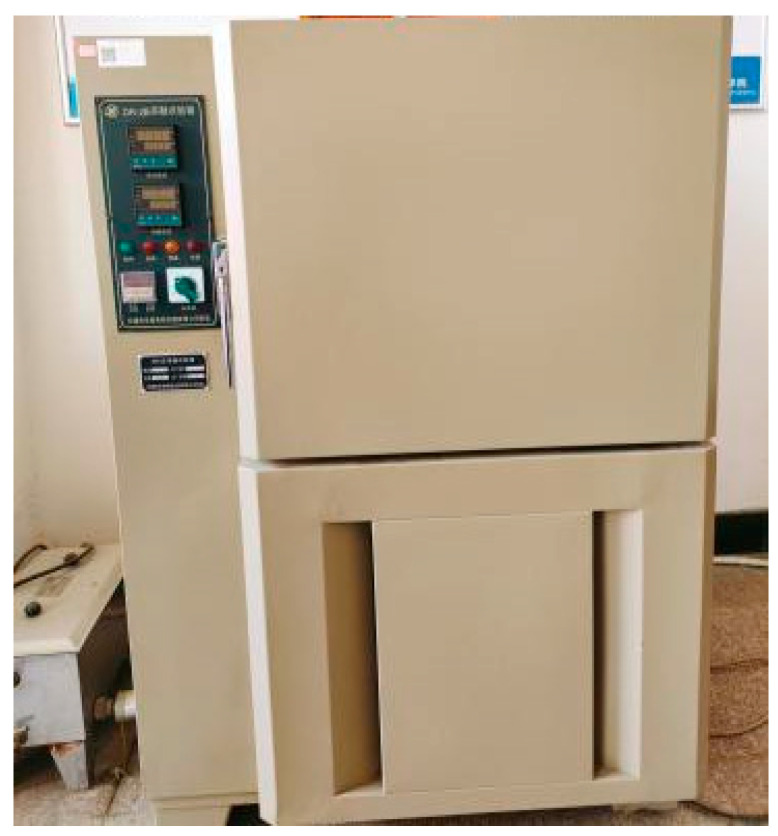
Freeze–thaw test chamber.

**Figure 2 materials-15-03856-f002:**
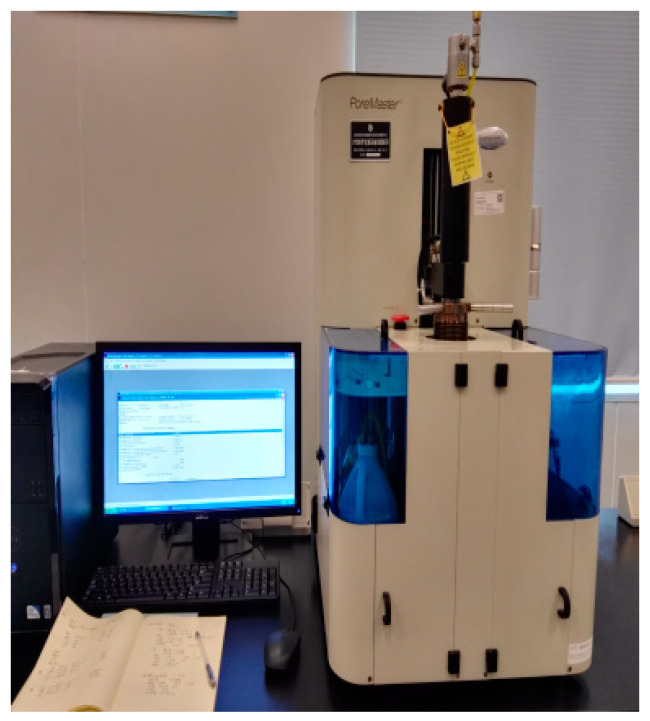
Automatic mercury injection instrument.

**Figure 3 materials-15-03856-f003:**
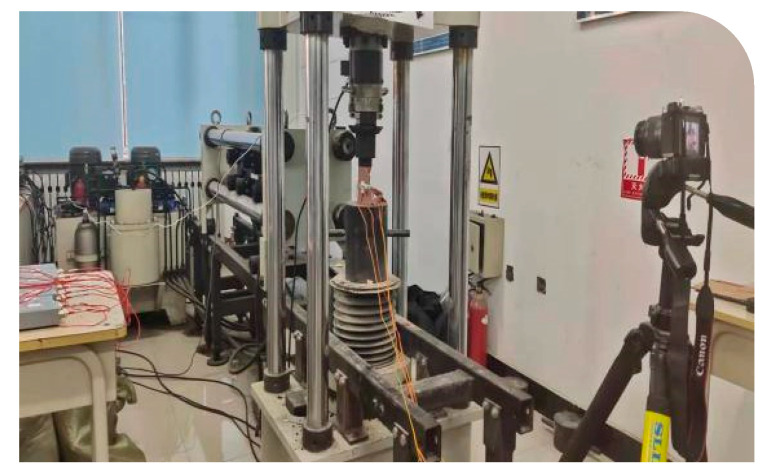
Rock triaxial test system.

**Figure 4 materials-15-03856-f004:**
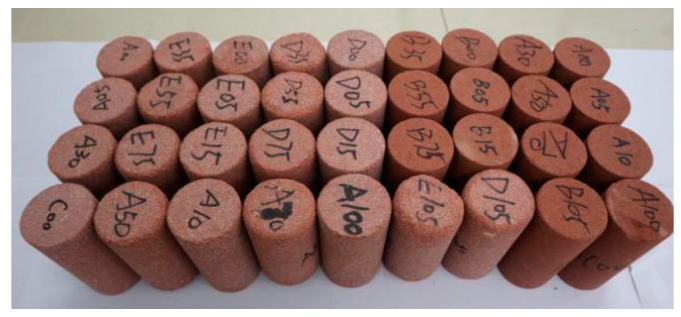
Red sandstone samples.

**Figure 5 materials-15-03856-f005:**
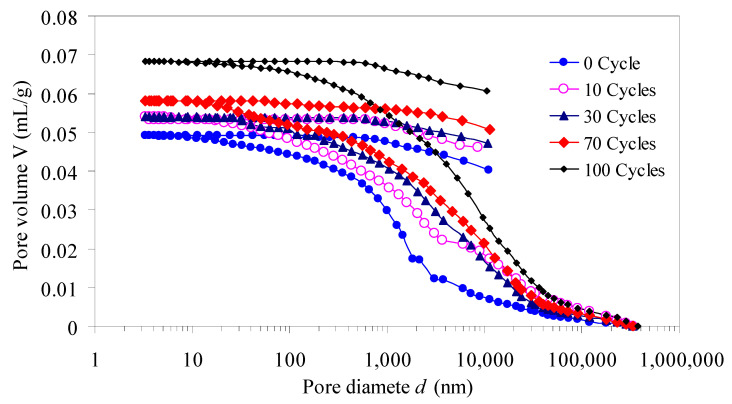
Pore volume versus pore diameter curves of red sandstone.

**Figure 6 materials-15-03856-f006:**
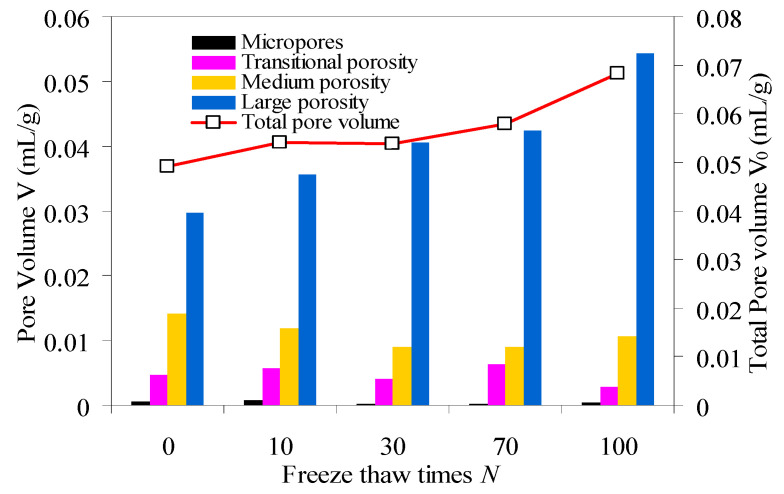
Pore volume distribution of red sandstone.

**Figure 7 materials-15-03856-f007:**
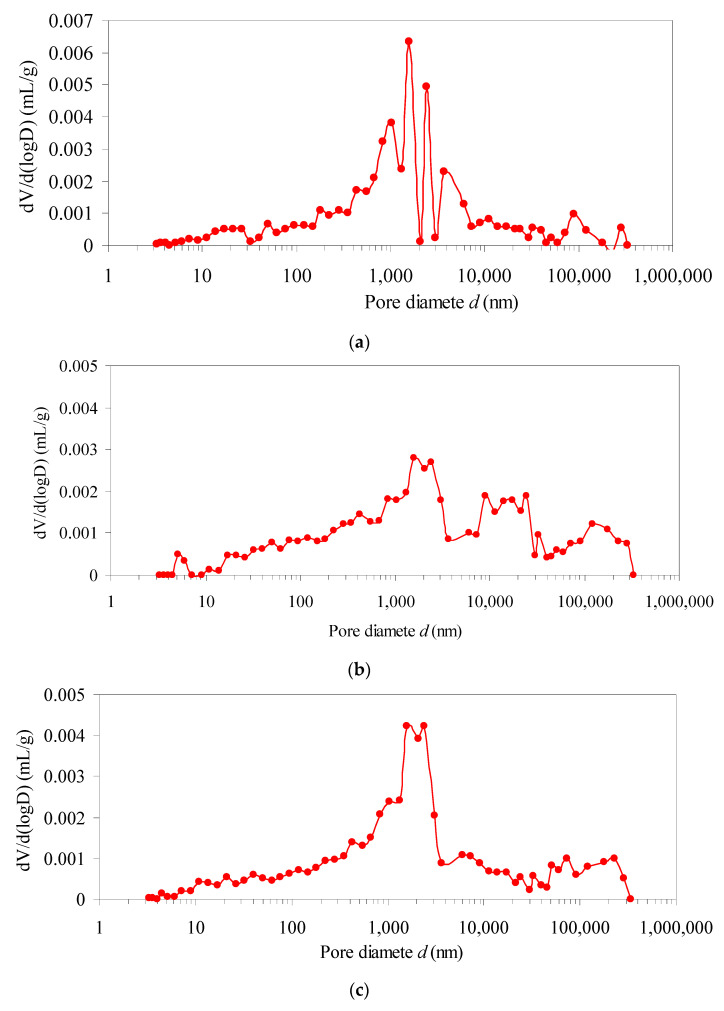
Pore size distribution. (**a**) 0 cycle; (**b**) 10 cycles; (**c**) 30 cycles; (**d**) 70 cycles; and (**e**) 100 cycles.

**Figure 8 materials-15-03856-f008:**
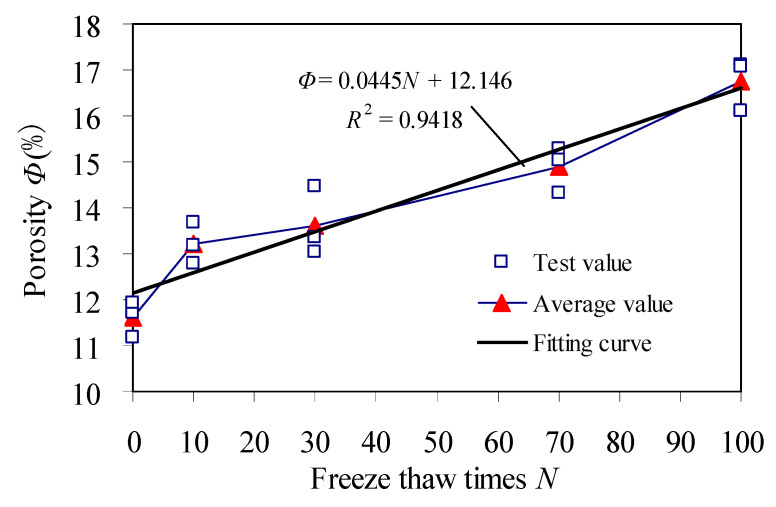
Porosity of red sandstone.

**Figure 9 materials-15-03856-f009:**
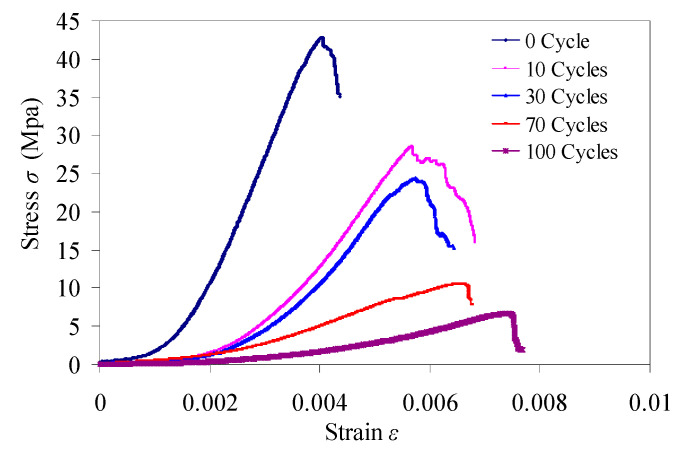
Typical stress–strain curves of samples.

**Figure 10 materials-15-03856-f010:**
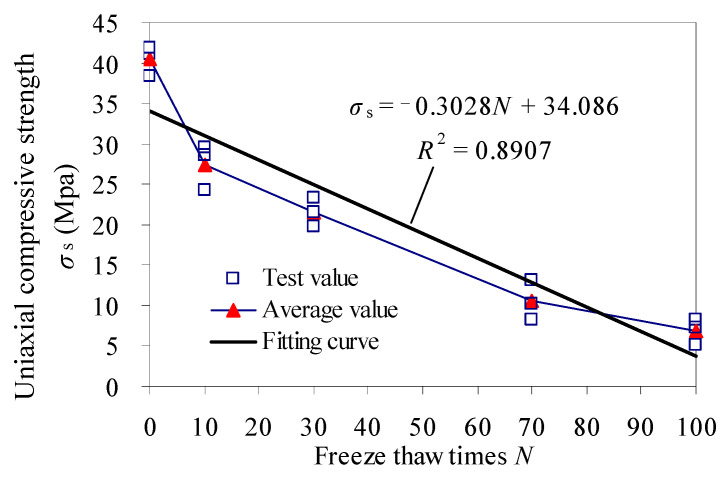
Uniaxial compressive strength of samples.

**Figure 11 materials-15-03856-f011:**
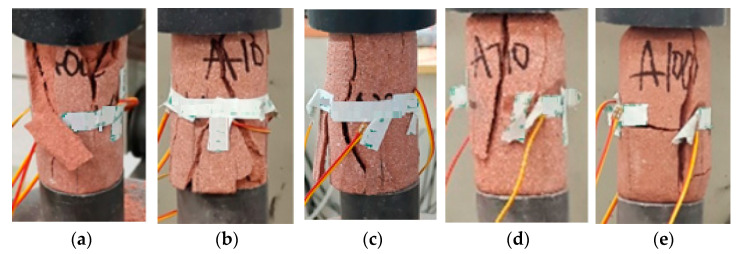
Failure modes of samples. (**a**) 0 cycle; (**b**) 10 cycles; (**c**) 30 cycles; (**d**) 70 cycles; and (**e**) 100 cycles.

**Figure 12 materials-15-03856-f012:**
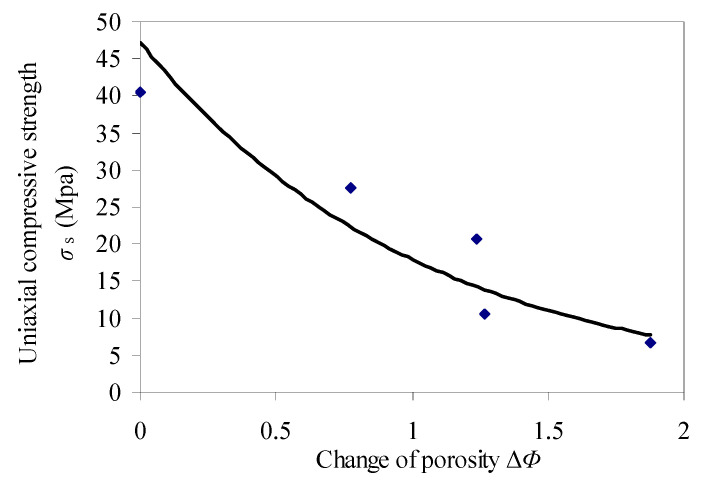
Change of porosity versus compressive strength.

**Figure 13 materials-15-03856-f013:**
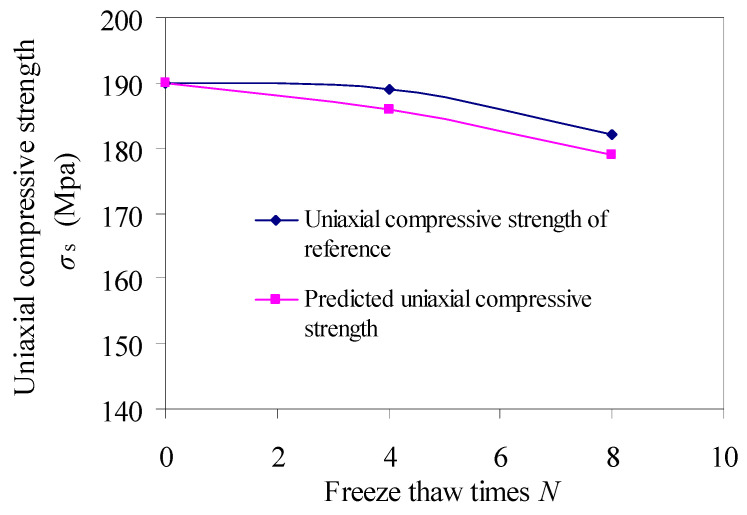
Uniaxial compressive strength of limestone.

**Table 1 materials-15-03856-t001:** Pore volume parameters of red sandstone.

Freeze–Thaw Cycles	Volume of Different-Sized Pores (cm^3^/g)				Total Pore Volume (cm^3^/g)
	Micropores	Transitional-Pores	Mesopores	Macropores	
0 cycle (No Freeze–thaw)	0.00071	0.00469	0.01406	0.02963	0.04909
10 cycles	0.00082	0.00580	0.01157	0.03545	0.05364
30 cycles	0.00015	0.00401	0.00896	0.04059	0.05371
70 cycles	0.00027	0.00643	0.00905	0.04231	0.05806
100 cycles	0.00043	0.00287	0.01074	0.05426	0.06830

**Table 2 materials-15-03856-t002:** Porosity of red sandstone.

Freeze–Thaw Cycles	Test Value of Porosity (%)			Average Value of Porosity (%)
0 cycle (No Freeze–thaw)	11.1879	11.7110	11.9214	11.6068
10 cycles	13.6653	12.8019	13.1707	13.2126
30 cycles	13.0427	14.4541	13.3515	13.6161
70 cycles	14.3066	15.2926	15.0426	14.8806
100 cycles	17.0975	16.1235	17.0558	16.7589

**Table 3 materials-15-03856-t003:** Uniaxial compressive strength of samples.

Freeze–Thaw Cycles	Test Value of Uniaxial Compressive Strength (MPa)			Average Value of Uniaxial Compressive Strength (MPa)
0 cycle (No Freeze–thaw)	41.082	41.917	38.271	40.423
10 cycles	29.472	24.332	28.657	27.487
30 cycles	23.350	21.515	19.795	21.553
70 cycles	13.104	8.246	10.247	10.532
100 cycles	8.283	5.123	7.150	6.852

## Data Availability

Data sharing is not applicable to this article.

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
