# Peer review of "Pore Structure Characteristics and Strength Variation of Red Sandstone under Freeze–Thaw Cycles"

_materials, 2022, doi:10.3390/ma15113856_

Round 1

Reviewer 1 Report

The abstract is too long and requires modification. 

Authors should focus on the objective of the article

Authors should write the aim of the research at the end of the introduction

Authors should explain the relation between freeze-thaw cycles and porosity.

It is better if the authors can explain the effect of porosity on the compressive strength.

Results and discussion are not supported by relevant references.

The authors should improve the language, possibly through the help of a native English speaker.

The conclusion is too long; the authors should summarize the conclusion and focus only on the best results.

Author Response

Dear reviewer,

Thank you very much for your valuable comments. According to your suggestions, we have revised them one by one. If there is anything inappropriate for me to modify, I hope to give me the opportunity to modify my paper next time.

Thank you again for your valuable comments on the paper.

Sincerely

Gao Hongmei

Reviewer 2 Report

The topic of “Pore structure characteristics and strength of saturated red sandstone under

freeze-thaw cycles” is interested, however, there is major correction should be done before submission:

abstract: needs to be reconstructed

  1. please start with the current missing and the problem related saturated red sandstone to in the present literature.
  2. The whole abstract is qualitative and there is no any numerical value to represent the results. Please make your abstract a quantitives as well.
  3. The last sentence of theoretical should be removed as the theoretical part is not clear. You can just highlight the regression analysis of your data only.

Introduction: needs to be revised

  1. Please show us the previous works that is related to saturated red sandstone.
  2. The authors stated that “However, there are few quantitative research results on the relationship between pore structure characteristics and compressive strength in rocks under freeze-thaw cycles”. Where is the references. What type of rock in this sentence?

Methodology

  1. Please write all references of standard and specification that are related your experimental works.

Results:

  1. Strength prediction model of red sandstone:

This section must be checked as it is not correct and not followed mathematics. How they develop equation without any boundary or initial condition or any information.

conclusion: include numerical values

Author Response

(The authors gave the same response as above.)

Reviewer 3 Report

Dear Authors, 

The study investigated the freeze-thaw effects of red sandstone experimentally. The study is an experimental study and is interesting as a subject.  I believe that it can be published after the necessary corrections are made in the article.

  • It is beneficial for the authors to use short and understandable sentences rather than long sentences. In addition, the language of the publication should be checked once again by the authors.
  • In the abstract, information about the samples used and the experimental study should be added in 1-2 sentences.
  • There are current references on this subject and materials. Some suggested references;

Huang, S., et al., (2022). Pore structure change and physico-mechanical properties deterioration of sandstone suffering freeze-thaw actions.Construction and Building Materials, 330, 127200.

Jia, H., et al. (2020). Evolution in sandstone pore structures with freeze-thaw cycling and interpretation of damage mechanisms in saturated porous rocks.Catena, 195, 104915.

Karasin, A., et al. (2022). The Effect of Basalt Aggregates and Mineral Admixtures on the Mechanical Properties of Concrete Exposed to Sulphate Attacks. Materials,15(4), 1581.

and similar studies. 

  • It would be beneficial to give the literature part in detail together with current references.
  • At the end of the introduction, what has been done in the article, the importance and novelty of the article should be revealed with clear sentences.
  • Comparison with results from other studies is recommended.
  • The samples considered in the study and their properties can be shown in a table.
  • It is useful in expanding the conclusion part.

Kind regards

Author Response

(The authors gave the same response as above.)

Reviewer 4 Report

In the Reviewer opinion the research paper entitled “Pore structure characteristics and strength of saturated red sandstone under freeze-thaw cycles” is average.

Research shows that: with the increase of freeze-thaw cycles, cohesion of mineral particles in red sandstone decreases and the weak surfaces of cementation weaken under the repeated action of frost heaving force.With the increase of freeze-thaw cycles, freezing-thawing damage continues to accumulate, uniaxial compressive strength decreases gradually, and the crack propagation direction changes from axial through-through failure mode to transverse and axial simultaneous failure mode. Taken the change of porosity as a parameter, the functional relationship between uniaxial compressive strength and the change of porosity in red sandstone is established.

Some comments which greatly enhance the understanding of the paper and its value are presented below. Specific issues that require further consideration are:

  1. The title of the manuscript is matched to its content.
  2. The structure of the manuscript is rather proper.
  3. The Introduction  covers the cases.
  4. In the Reviewer’s opinion, the current state of knowledge relating to the manuscript topic has not been covered and clearly presented.
  5. An analysis of the manuscript content and the References shows that the manuscript under review constitutes a summary of the Author(s) achievements in the field.
  6. Please corrected all drawings – add name and units of each axis.
  7. Article has flaws, additional experiments needed, research not conducted correctly.
  8. In the Reviewer’s opinion, the bibliography, comprising 31 references, is not representative and exhaustive.
  9. I suggest expanding the conclusions.
  10. In the Reviewer’s opinion the manuscript can be published in the journal, but after major revision.

Author Response

(The authors gave the same response as above.)

Round 2

Reviewer 1 Report

No comments

Author Response

Dear Reviewer,

Thank you for your comments and comments that are very helpful to the improvement of this article.

Best wishes to you.

Sincerely

Gao Hongmei

Reviewer 2 Report

Dear author. Thanks for doing most of the comments.
regarding the mathematics part, you did not consider my idea.
your paper can be accepted if the equations will be removed as it is not correct.
for example, your equations 1 to 4 are just repeated equations and simple as well as a general equations. the equations aimed to say that the change of porosity with respect to N is equal to a. what is a? it is constant?
is it nonlinear?
what is the initial condition?
check your curve carefully?

your curve (Fig. 12) is developed from regression analysis. so that no need to show us these equations. 

Author Response

(The authors gave the same response as above.)

Reviewer 4 Report

Authors corrected manuscript follow to Reviewers suggestion. In my opinion can be published in the Journal.

Author Response

(The authors gave the same response as above.)
